# Temporal analysis of genetic diversity and gene flow in the threatened catfish *Pseudoplatystoma magdaleniatum* from a dammed neotropical river

Kevin León García-Castro⌖, Edna Judith Márquez ⓘ*⌖

Facultad de Ciencias Universidad Nacional de Colombia–Sede Medellín, Laboratorio de Biología Molecular y Celular, Escuela de Biociencias, Medellín, Antioquia, Colombia

⌖ These authors contributed equally to this work.
* ejmarque@unal.edu.co

**Data Availability Statement:** All relevant data are within the manuscript and its Supporting Information files.

## Abstract

The striped catfish *Pseudoplatystoma magdaleniatum* is a large-sized migratory species from the north Andes region, endemic to Magdalena basin and one of the major fishery resources. Despite the estimated reduction of over 80% of the fisheries production of this species throughout the basin in recent decades, its population in the lower Magdalena-Cauca basin showed healthy genetics after molecular analyses. However, the current conservation status of this species and several habitat disturbances demand the re-evaluation of its population genetics to infer evolutionary risks and assess potential changes. This work analyzed a total of 164 samples from the Cauca River collected downstream the Ituango Dam between 2019–2021 using species-specific microsatellite markers to compare the genetic diversity and structure in samples collected between 2010–2014 from the lower Magdalena-Cauca basin, previously analyzed. Our results showed a relatively stable panmictic population over time (4 to 10 years), with high genetic diversity and evidence of recent bottleneck. Promoting habitat connectivity to conserve gene flow, characterizing diversity and genetic structure over the entire basin, and integrating the results with future monitoring are important aspects for the management planning for *P. magdaleniatum* in the Magdalena-Cauca basin.

## Introduction

The Cauca River is the main tributary of the Magdalena River Basin in the north Andes region, subject to strong anthropogenic pressures because of the settlement of a large part of the Colombian population and the impact of the local economy [1]. The greatest threats to the Magdalena-Cauca basin are mainly related to mining, water pollution and habitat modification by agriculture, livestock production, deforestation and construction of dams [1,2]. These and other factors, such as overfishing, introduction of exotic species and genetic contamination of populations [3], are of growing concern due to their potential impacts on biodiversity.

**Funding:** This study was supported by a grant framed under the Project "Variabilidad genética de un banco de peces de los sectores medio y bajo del río Cauca" (CT-2019-000661, Empresas Públicas de Medellín and Universidad Nacional de Colombia, Sede Medellín). Funders do not play any role in the study design, data collection and analysis, decision to publish, or preparation of the manuscript.

**Competing interests:** The authors have declared that no competing interests exist.

In particular, the Magdalena River basin has 233 fish species (14.5% of the freshwater ichthyofauna of the country), 68.1% of which are endemic and more than 15% are included on the Red List of freshwater fishes of Colombia with some degree of threat [4,5].

One of these endemic species is the striped catfish *Pseudoplatystoma magdaleniatum*, a large migratory fish and the most important fishery resource of Colombia after bocachico *Prochilodus magdalenae*, with 11% of the total landings in the Magdalena basin in 2019 [6]. However, it is estimated that its fishery production has been reduced by around 90% since 1970, which is the main cause of *P. magdaleniatum* being classified as Endangered or Critically Endangered on national and international red lists [5,7]. The population of this species in the middle and lower sectors of the Magdalena-Cauca basin showed high genetic diversity and absence of population structure [8]. Other migratory species of high commercial interest, such as *Pimelodus yuma* (nicuro), *Pimelodus grosskopfii* (barbudo) and *Prochilodous magdalenae* (bocachico) not only showed gene flow in the middle and lower sectors of the Cauca River but also high degree of inbreeding [3].

The fragmentation of rivers by dam constructions modifies natural landscape and potentially impacts migratory rheophilic species [9–12]. The hydroelectric project Hidroituango in the Cauca River gives rise to multiple factors that potentially threaten the biodiversity of this basin [2,3,13,14]. This dam (here called the Ituango Dam) is in the Cauca River canyon, an area of rapids and slopes that naturally limit the upstream migration of species such as *P. magdaleniatum*, so populations of this fish are mainly found downstream of this hydroelectric site [4,15]. Although the migratory route of *P. magdaleniatum* is not considered interrupted, potential downstream effects may disturb the migration behavior of this species and might have impacts at population level, as migration is a key factor in the evolutionary processes of wild populations [16–18]. For instance, sediment retention upstream can alter the balance in sedimentation levels and reduce the availability of nutrients downstream [19,20], especially in floodplains, which are crucial places for completing the life cycle of migratory fishes in the basin [14]. Similarly, modification of water flow due to regulation of discharges can alter the endocrine response associated with the reproductive migration of fish downstream, as has been reported in *P. magdalenae* from La Miel River [21]. Other factors such as organic mercury levels increase and pH and oxygen alteration downstream, because of trophic and biochemical effects that occur mainly upstream, are more particular impacts that depend on the characteristics of the dam and the river system [22,23].

Knowledge about real impacts of such factors on population genetics of non-fragmented species is quite limited. The few studies that report some degree of genetic change on a temporal scale in populations located downstream of the dam, constitute cases where migratory routes are effectively fragmented and those changes are generally associated with a reduced effective population size or bottleneck events that might be related to that isolation (*e. g.* [24–27]). Further, considering the multiple threats to biodiversity in the Magdalena-Cauca basin, genetic monitoring emerges as an approach to estimate the threat status and potential changes of wild populations over time, mainly of those species with some degree of conservation concern and subjected to disturbances in its ecosystem [28,29]. Therefore, this study analyzed the population genetics of *P. magdaleniatum* on a temporal scale, using samples collected in the Magdalena-Cauca basin before and after the construction of the Ituango Dam, using species-specific microsatellite markers.

The expectations of this work were to find high genetic diversity and no population structuring in *P. magdaleniatum* of the Cauca River, based on the findings previously reported by García-Castro *et al.* [8]. Additionally, despite the genetic evidence of a reduced population size [8] and the persistence of different anthropogenic pressures on this species [2], substantial genetic changes in terms of diversity or structure in the population of *P. magdaleniatum* over the considered time scale were expected. These two hypotheses are mainly based on: (i) the

absence of fragmentation of the population of *P. magdaleniatum* in the Cauca River, (ii) the short period of time separating samplings before and after the dam construction in relation to the generation length of this species (approximately four years; see [7]), and (iii) the delay that may exist between contemporary demographic/environmental processes and the expression of their impact on genetic diversity and structure [30,31].

## Materials and methods

### Sampling and genotyping

The Cauca River runs along 1,350 km from south to north in the Colombian Andes mountains, from the Alto Cauca valley between 3,200 and 1,000 meters above sea level to the Cauca canyon in the middle and lower sectors of the basin. The geography presents natural barriers for many migratory species such as *P. magdaleniatum* and where the current Ituango dam of the Hidroituango hydroelectric project is located. In the last 500 km, the Cauca River receives important tributaries such as the Ituango and Nechí rivers and feeds several floodplains during rainy periods until it joins the Magdalena River in the Mompós Depression. This last sector has an important part of the fish biodiversity of the Magdalena-Cauca basin and concentrates the most commercially important fishery resources [4,32].

This study analyzed a total of 164 muscle or fin tissues from individuals of *P. magdaleniatum* collected between the years 2019–2021 (Ex-post sample), in seven sectors of the middle and lower part of the Cauca River (S2, S3, S4, S5, S6, S7 and S8; see [33]), located downstream of the Ituango Dam. These samples were supplied by Grupo de Ictiología from Universidad de Antioquia (GIUA), Fundación Humedales and Grupo de Biotecnología Animal from Universidad Nacional de Colombia, Sede Medellín. Due to low and heterogeneous sample numbers among sectors, the samples were grouped into three larger sectors denoted as: S2-S3 ($N = 49$), S4-S5 ($N = 59$) and S6-S7-S8 ($N = 56$), for population genetic analyzes (Fig 1).

DNA extraction was performed using the commercial GeneJet Genomic DNA Purification Kit (Thermo Scientific). The microsatellite regions were amplified using 13 primer pairs previously designed and evaluated in wild populations of *P. magdaleniatum* (Psm03, Psm04, Psm06, Psm11, Psm14, Psm16, Psm18, Psm19, Psm21, Psm22, Psm24, Psm25 and Psm26) using PCR conditions reported by García-Castro *et al.* [8]. The amplified fragments were separated by capillary electrophoresis on an ABI 3730 XL automatic sequencer (Applied Biosystems), using LIZ600 (Applied Biosystems) as an internal molecular size marker. Alleles were recorded in GeneMarker software v.3.0.0 and amplification and scoring errors were evaluated in Micro-Checker v.2.2.3 [34].

### Genetic diversity and demographic events

To estimate the genetic diversity of the Ex-post sample, the average number of alleles per locus ($Na$), allelic range ($Ra$) and the expected ($H_E$) and observed ($H_O$) heterozygosities were calculated using the GenAlEx v6.51b2 program [35,36] and the allelic richness ($Ar$) was calculated in FSTAT v.2.9.4 [37]. Inbreeding coefficients ($F_{IS}$) and deviations from Hardy-Weinberg (HWE) and linkage (LD) equilibria were evaluated in Arlequin v3.5.2.2 [38]. The multilocus significance values for the HWE by population or sector were calculated using the Fisher's Exact test integrated in the web version of GENEPOP v4.7.5 [39,40].

Drastic reduction in population size (bottleneck) was evaluated using the excess heterozygosity test within BOTTLENECK v1.2.02 [41] using its default parameters, comparing three likely mutation models for microsatellites [42]: the infinite alleles model (IAM), stepwise mutation model (SMM) and two-phase mutation model (TPM). A second approach consisted of calculating the standardized *M* index of Garza and Williamson [43] in Arlequin v3.5.2.2

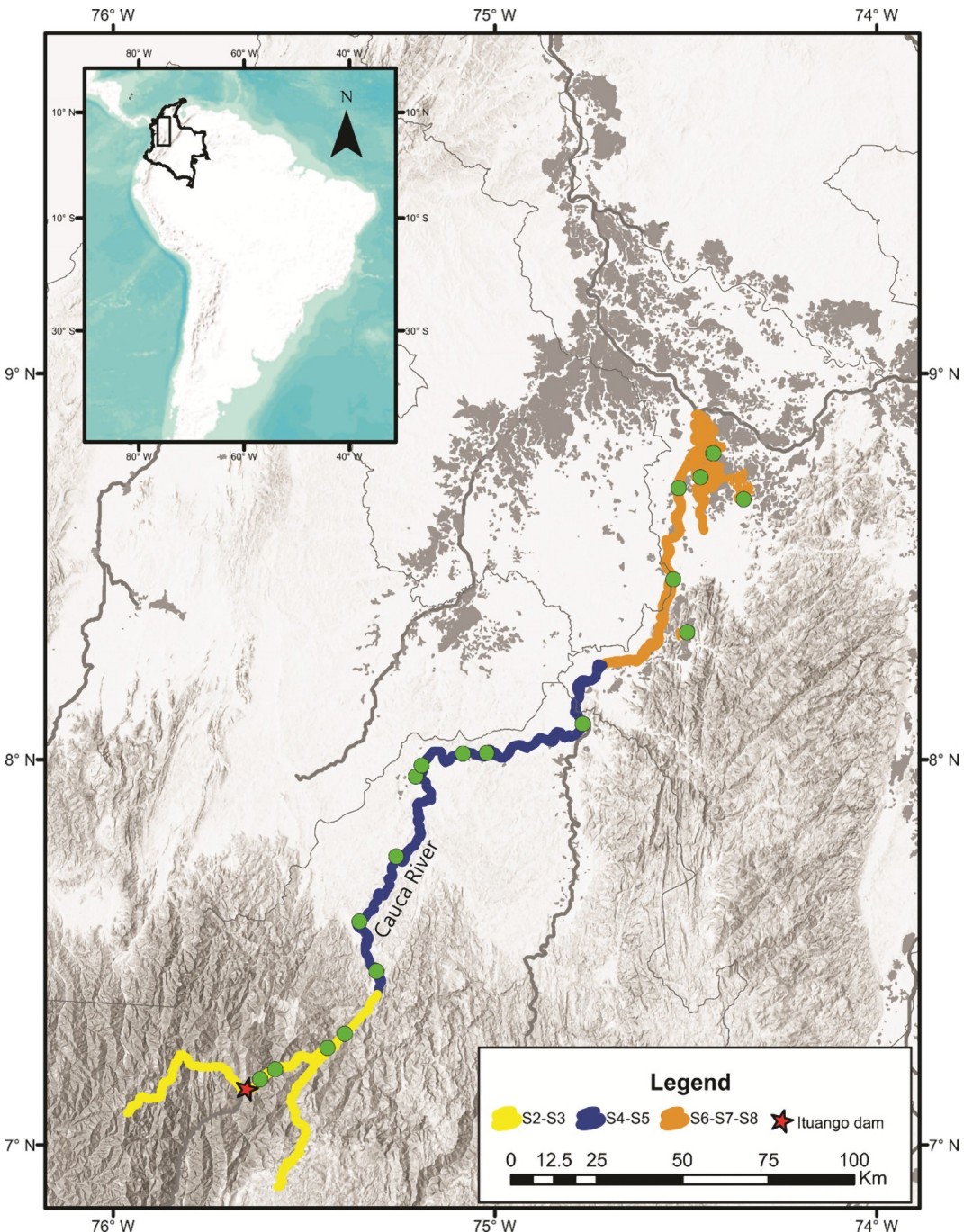

**Fig 1. Sampling sites (circles) of *Pseudoplatystoma magdaleniatum* in three sectors of the Cauca River downstream the Ituango dam.** S2-S3: Ituango River mouth, Golondrina, Espíritu Santo River mouth, Puerto Valdivia; S4-S5: Puerto Jardín, Man River, El Doce, Cáceres, Caucasia, La Ilusión, Nechí, Palomar; S6-S7-S8: Guaranda, Tres Cruces and floodplains La Raya, La Panela, Piqué and El Floral.

[38], which quantifies the reduction in the number of alleles with respect to the allelic size range of a population to detect recent bottleneck events. Additionally, effective population size (*Ne*) was estimated in NeEstimator v2.1 [44] using LD and evaluating allele frequencies greater than or equal to 0.02, since lower allelic frequencies tend to overestimate *Ne* [44,45].

Finally, the most likely first-generation migrants (individuals from another sampled sector) analysis was performed using GENECLASS2 software [46], implementing a Bayesian method [47] and the unbiased Monte Carlo resampling method [48], with 13 loci, 10,000 individuals and a significance level of 0.01. In addition, for multiple comparisons analyzes, the Bonferroni correction was applied.

## Genetic structure

For the genetic structure analysis in the Ex-post sample, pairwise comparisons of the standardized indices $F'_{ST}$ [49,50] and Jost's $D_{EST}$ [50,51] were used, and an analysis of molecular variance (AMOVA; [46]) was performed, using GenAlEx v6.51b2 [35,36]. Genetic differentiation between sectors was evaluated through a discriminant analysis of principal components (DAPC) using the R package Adegenet [52]. Finally, a Bayesian clustering analysis was performed using Structure v2.3.4 [53] using 800,000 Markov chain Monte Carlo (MCMC), 80,000 of these as burn in, and the models LOCPRIOR, genetic mixture and correlated alleles. The results were evaluated at K = 1 to K = m + 3, where m is the number of *a priori* populations [54], with 20 repeats each. Then, to determine the most likely number of populations (K), StructureSelector software [55] was used to calculate six statistics (*MedMeaK, MaxMeaK, MedMedK* and *MaxMedK*: [56]; *Ln Pr (X | K)*: [53]; and *ΔK*: [53]) and to plot the histogram of co-ancestry probabilities of all individuals.

## Comparative analysis of temporal samples

To explore genetic changes on a temporal scale for population of this species, genotypes previously obtained for individuals collected during the years 2010–2014 (Ex-ante sample) in sectors of the middle and lower sites of the Magdalena-Cauca basin, using the same set of microsatellite loci and the same methodology for genotyping them (see [8]), were included in the analysis, totaling of 311 individuals of *P. magdaleniatum*. The genetic diversity (*Na, Ar, Ra, $H_E$, $H_O$, $F_{IS}$*, HWE), demographic events (evaluation of bottlenecks and *Ne* estimation) and genetic structure between the Ex-ante and Ex-post samples of *P. magdaleniatum* were evaluated using the above methodology and descriptive comparisons. Additionally, a genic differentiation test (G test) was carried out using the web version of GENEPOP v4.7.5 [39,40], which evaluate the pairwise differences of the allelic distribution at each locus between the temporal samples, using the modified Fisher Exact Test to evaluate statistical significance.

Finally, the *Ne* was estimated using the temporal method (two or more temporally separated samples) in NeEstimator v2.1 [44], which was applied in addition to LD (single sample). For this procedure, a minimum separation of two generations between samples was assumed based on a generation length of about four years for *P. magdaleniatum* [7]. Only the *Fs* value was considered as an estimator of the change in allelic frequencies, which is less biased than its analogues *Fc* and *Fk*, although less precise [57]. Plan II was considered as the sampling method, which does not require knowing the population size (*N*) as a parameter for the estimation of *Ne* and assumes that individuals are collected before reproducing and without returning to the original population [44].

# Results

## Genetic diversity

Genotyping errors due to segregation of null alleles were not detected, and stuttering effects were corrected when present based on Micro-Checker suggestions. Genetic diversity in the three sectors was high, with similar values of average numbers of alleles per locus (*Na*), allelic

**Table 1. Average values per locus of genetic diversity metrics for *Pseudoplatystoma magdaleniatum* in the Ex-post (collected between years 2019–2021 in three sectors of the Cauca River downstream of the Ituango dam) and Ex-ante (years 2010–2014; [8]) samples.**

| Sample | N | Na | Ar | Ra | $H_O$ | $H_E$ | P | $F_{IS}$ |
|---|---|---|---|---|---|---|---|---|
| S2-S3 | 49 | 9.154 | 8.820 | 36.308 | 0.753 | 0.771 | 0.249 | 0.004 |
| S4-S5 | 59 | 10.154 | 9.461 | 41.231 | 0.776 | 0.785 | **0.042** | 0.003 |
| S6-S7-S8 | 56 | 8.846 | 8.497 | 34.769 | 0.756 | 0.771 | **0.023** | 0.021 |
| Overall (Ex-post) | 164 | 11.000 | 10.762 | 43.077 | 0.762 | 0.777 | **0.007** | 0.023 |
| Ex-ante | 147 | 11.308 | 11.162 | 44.615 | 0.765 | 0.783 | 0.094 | -0.019 |

*N*: Sample size, *Na*: Number of alleles, *Ar*: Allelic richness, *Ra*: Allelic range, $H_E$: Expected heterozygosity and $H_O$: Observed heterozygosity, $F_{IS}$: Inbreeding coefficient, *P*: p-value of the Hardy Weinberg equilibrium test. Values in bold denote statistical significance.

richness (*Ar*), allelic range (*Ra*) and expected (*$H_E$*) and observed (*$H_O$*) heterozygosities (Table 1). Inbreeding coefficients (*$F_{IS}$*) were positive (0.003–0.021), but not significant (P> 0.05).

Furthermore, it is worth noting that apart from Psm06 in a single sector, no locus exhibited departure from Hardy-Weinberg equilibrium across population (S1 Table), therefore, the significant values across loci observed in sectors S4-S5 and S6-S7-S8, and in the overall Ex-post sample (P-values of 0.042, 0.023, and 0.007, respectively), may be potentially biased by Fisher's Exact test.

At temporal scale, the Ex-post sample showed slightly reduced genetic diversity (*Na*, *Ar*, *Ra*, *$H_O$* and *$H_E$*) respect to the Ex-ante sample, as well as an increased inbreeding coefficient, although this latter was non-significant in both temporal samples (Table 1). Indeed, only one locus (Psm24) showed differences in allelic distributions between the two samples (*P* = 0.012), so that in general, the allelic distributions did not evidence variations over time (*P* multilocus = 0.077).

## Demographic events

The result of the excess heterozygosity test to detect recent bottleneck events was significant (*P* <0.017) in the three sectors and overall (Ex-post) using the IAM, whereas in the other models were non-significant. Additionally, the *M* indices were lower than the reference value of 0.68 [43]. These results together indicate a recent reduction in the population size of *P. magdaleniatum* in the Cauca River, same as the Ex-ante sample (Table 2).

Moreover, *Ne* using LD varied between 814.9 and ∞ in the three sectors. Estimates equal to ∞ can be fully explained by a sampling error that is greater than the genetic drift signal, so

**Table 2. Assessment of recent bottleneck events and estimation of effective population size in Ex-post (2019–2021, sectors S1—S8 of the Cauca River) and Ex-ante (years 2010–2014; [8]) samples of *Pseudoplatystoma magdaleniatum*.**

| Sample | M | IAM | TPM | SMM | Ne | CI |
|---|---|---|---|---|---|---|
| S2-S3 | 0.21 | **0.001** | 0.207 | 0.863 | 814.9 | 223.7 - ∞ |
| S4-S5 | 0.23 | **0.000** | 0.108 | 0.936 | ∞ | 1,008.2 - ∞ |
| S6-S7-S8 | 0.21 | **0.001** | 0.122 | 0.554 | ∞ | 695.7 - ∞ |
| Overall (Ex-post) | 0.23 | **0.000** | 0.095 | 0.996 | 2,126.3 | 626.9 - ∞ |
| Ex-ante | 0.23 | **0.000** | 0.137 | 0.998 | 1,379.4 | 414.4 - ∞ |

*M*: Standardized index of Garza-Williamson [43]. IAM, TPM, SMM: p-values (in bold for statistical significance after Bonferroni correction) of the heterozygosity excess test implemented in BOTTLENECK v1.2.02 [41] according to the assumed mutational model. *Ne*: Effective population size (*Ne*) estimated using the Linkage disequilibrium (LD) method. CI: Confidence interval of the *Ne*.

**Table 3. Pairwise comparison of the standardized genetic structure indices $F'_{ST}$ (below the diagonal) and Jost's $D_{EST}$ (above the diagonal) in the Ex-post sample (collected between 2019–2021, sectors S1—S8 in the Cauca River).**

| Sitio | S2-S3 | S4-S5 | S6-S7-S8 |
|---|---|---|---|
| S2-S3 | - | 0.001 | 0.008 |
| S4-S5 | 0.012 | - | 0.007 |
| S6-S7-S8 | 0.017 | 0.012 | - |

No statistical significance was found after correction for multiple comparisons.

they do not provide evidence that the population is very large [45]. This method showed that the overall estimation (Ex-post) was higher than that of the Ex-ante sample. These two values were close to that obtained using the temporal method: 1,745.3 (CI: 446.2 –∞). Therefore, both methods suggest that the *Ne* of *P. magdaleniatum* is greater than 1,300 in both temporal samples, noting that confidence intervals overlap, being the lowest limit value of 414 (Table 2).

Finally, the exploration of migratory events among sectors of the Ex-post sample using the GENECLASS2 software [46], detected three individuals collected in S4-S5 as likely immigrants (P: 0.006, 0.005 and 0.009) from the lowest sector (S6-S7-S8), whereas another individual collected in the highest sector (S2-S3) was assigned to the lowest sector S6-S7-S8 ($P = 0.006$); supporting the gene flow throughout the studied area.

## Genetic structure

Although the overall structure index calculated using the AMOVA was significant ($F'_{ST\,[2,\,327]} = 0.014$; $P = 0.013$), geographical genetic structure for the Ex-post sample was fully explained by the variances among individuals (6%) and within individuals (94%). The absence of geographical genetic structure was supported by both $F'_{ST}$ and Jost's $D_{EST}$ indices (Table 3) and by the DAPC (Fig 2A), which showed an overlapping of the three sectors. Additionally, although the estimators of the best *K* from the Bayesian analysis differed, suggesting $K = 1$ (*Ln Pr (X | K)*), $K = 2$ (*MedMedK, MedMeanK, MaxMedK, MaxMeanK*) and $K = 5$ (*ΔK*), the homogeneous distribution of the co-ancestry probabilities along the sampling area when $2 \leq K \leq 3$ (Fig 2B and 2C) supports both the geographical genetic structure absence and a single genetic stock presence.

Finally, no differences were detected in the genetic structure when comparing the Ex-ante and Ex-post samples based on Jost's $D_{EST}$ ($D_{EST} = 0.001$, P = 0.351), the AMOVA (0% of the variance between samples), the DAPC (Fig 3A) and the Bayesian analysis (Fig 3B; $K = 1$ according to the estimators *Ln Pr (X | K)*, *MedMedK, MedMeanK, MaxMedK* and *MaxMeanK*), only differing in the $F'_{ST}$ index ($F'_{ST} = 0.009$, $P = 0.007$).

## Discussion

Factors supporting the need for ongoing genetic evaluation of wild populations are related to environmental and anthropogenic pressures that threaten the demographic and genetic viability of the species, particularly those recognized as having some degree of vulnerability [28,29]. In this work, 13 species-specific polymorphic loci previously developed and used to assess the population genetics of *P. magdaleniatum* were again used for genetic evaluation of 164 individuals (Ex-post sample, years 2019–2021) distributed in three sectors of the Cauca River downstream of the Ituango Dam.

The genetic diversity of the Ex-post sample was high, with $H_E$ values higher than both the average of Neotropical catfishes ($H_E$: 0.609 ± 0.210; [58]) and values reported in some

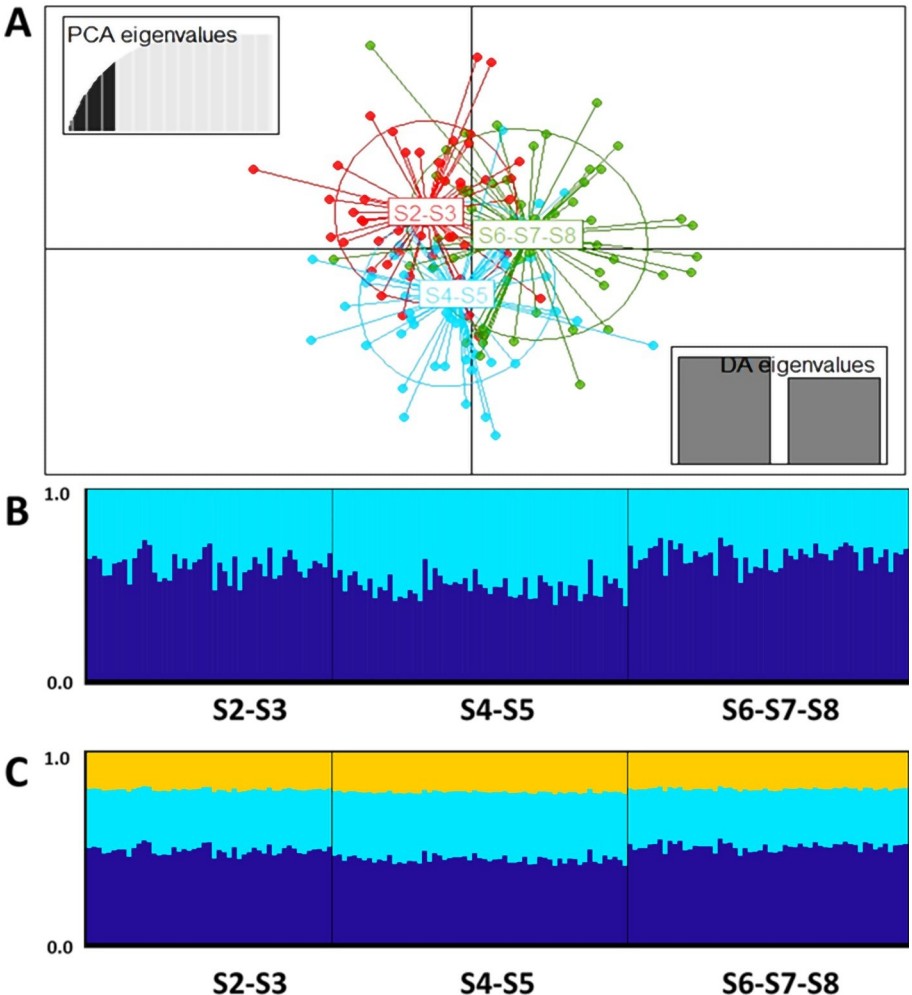

**Fig 2.** (A) Discriminant analysis of principal components (DAPC; 30 principal components retained and 71.3% of the variance) and co-ancestry probabilities when (B) $K = 2$ and (C) $K = 3$ of 164 individuals of *Pseudoplatystoma magdaleniatum* of the Ex-post sample (2019–2021, sectors S2 –S8 of the Cauca River).

populations of other species of the genus *Pseudoplatystoma* [59–64]. Additionally, this genetic diversity was similar to that of the Ex-ante sample (years 2010–2014) from the middle and lower sectors of the Magdalena-Cauca basin [8], supporting the initial hypothesis of this study about the stability of its genetic diversity over time. Since the comparisons of genetic diversity are mainly descriptive and no differences were detected in the allelic frequency distribution between the temporal samples (G test, P > 0.05), the observed differences may result from stochastic effects inherent to sampling [30,31].

An important indicator that measures the evolutionary potential of populations is the *Ne*, which can be estimated using a single sample over time (e.g., LD) or several samples separated by *n* generations (temporal method). Both approaches were performed in this work, finding an increase in *Ne* in the Ex-post sample using LD. Although some factors such as the presence of individuals of different generations (age structure) or high levels of immigration in the population can bias the estimation and could explain the observed differences between Ex-ante and Ex-post samples [65,66], such increasing in Ne remains uncertain mainly due to large

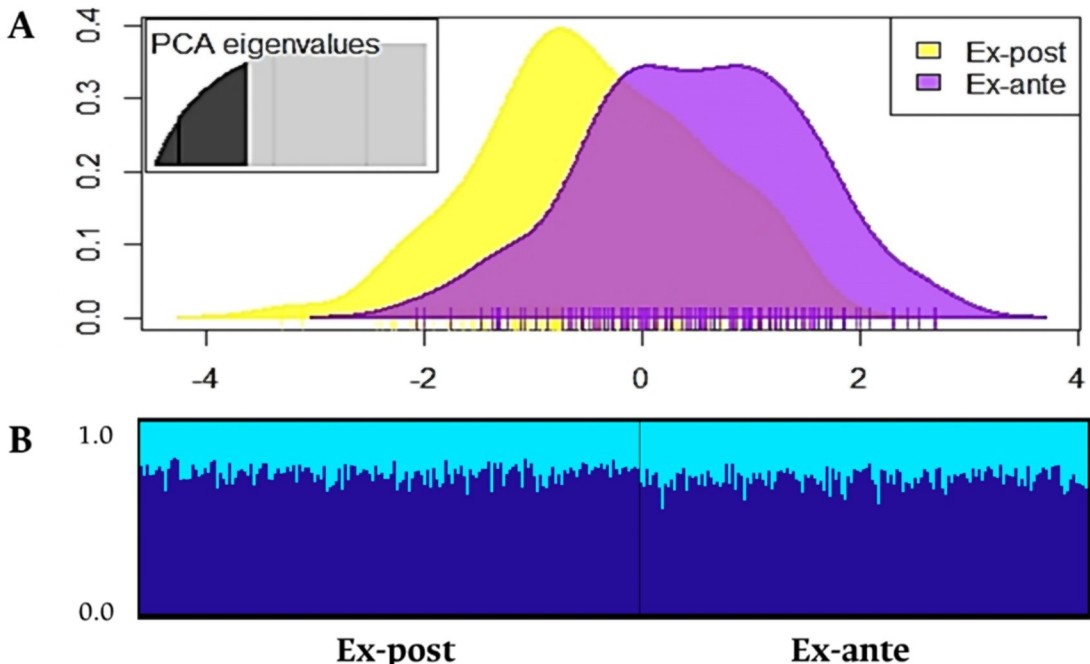

**Fig 3.** (A) Discriminant analysis of principal components (DAPC; 50 principal components retained and 85% of the variance) and (B) co-ancestry probabilities of 311 individuals of *Pseudoplatystoma magdaleniatum* (147 Ex-ante: Years 2010–2014, middle sectors and lower of the Magdalena-Cauca basin; 164 Ex-post: Years 2019–2021, sectors S1—S8 of the Cauca River).

confidence intervals and the imprecise estimation. However, these estimations were similar to those obtained using the temporal method, which generates a single value applicable to all the generations assumed between the samples. This is consistent with previous results using simulated data that demonstrate a similar performance of these two methods when it is assumed intervening few generations [45]. The most recent criteria established to consider that a wild population of a species can retain its evolutionary potential, is having values of $Ne \geq 1,000$ [67]. Although the estimates obtained in this study are higher than 1,300, the confidence intervals show the minimum values of 414, suggesting long-term evolutionary risks for the species [67], so this last value must be considered for support management measures to prevent genetic erosion of *P. magdaleniatum*, considering its current conservation status.

Moreover, our results show geographical genetic structure absence in the Ex-post sample and the presence of the same single genetic stock from the middle and lower sectors of the Magdalena-Cauca basin. This finding supports the prior hypothesis about the prevalence of a panmictic population (random reproduction) with high gene flow along the Cauca River in *P. magdaleniatum*. The absence of spatial barriers downstream of the Ituango Dam in the Cauca River and extensive floodplains that remain in the lower part of the Magdalena-Cauca basin would allow dispersal of this species and genetic connectivity of its population [1,68]. Although samples from the Magdalena River were not used in the Ex-post sample, it is likely that this species retains gene flow between these rivers (see [8]). This hypothesis should be tested in future studies that include samples from the Magdalena River, particularly of its upper sites and its tributaries such as the San Jorge River, which would allow testing whether this same genetic population is distributed throughout the rest of the basin.

In the wild populations, the number of generations required to detect genetic response to perturbations (time lag) could depend on several factors such as the mutation rate, dispersal

rate, population dynamics, the generation length of the species and *Ne* (see [30]). Therefore, a likely large dispersal rate, a large enough *Ne* and a short time frame evaluated in this work, translated into a small number of generations between the temporal samples, could explain the stability or the undetected changes in both genetic diversity and structure found in *P. magdaleniatum* that support the proposed initial hypotheses.

In conclusion, the results of this study indicate that *P. magdaleniatum* has high genetic diversity and is not genetically structured along the Cauca River, although preserving evidence of recent reductions in its population size. Given that reduced populations are more prone to the effects of genetic drift and therefore to changes in genetic diversity [43,45,69], monitoring of populations with evidence of bottleneck is relevant. On the other hand, the retention of dispersal and migratory behavior of this fish is crucial to maintain the gene flow found in its population, so ensuring the connectivity of the striped catfish habitat should be a priority in management plans over the long-term.

## Supporting information

**S1 Table. Hardy-Weinberg equilibrium test per locus and population.** P-values in bold (< 0.05) are significant.
(XLSX)

## Acknowledgments

We thank all of those who contributed somehow during lab work and improving the manuscript.

## Author Contributions

**Conceptualization:** Kevin León García-Castro, Edna Judith Márquez.

**Data curation:** Kevin León García-Castro, Edna Judith Márquez.

**Formal analysis:** Kevin León García-Castro, Edna Judith Márquez.

**Funding acquisition:** Edna Judith Márquez.

**Investigation:** Kevin León García-Castro, Edna Judith Márquez.

**Methodology:** Kevin León García-Castro, Edna Judith Márquez.

**Project administration:** Edna Judith Márquez.

**Resources:** Edna Judith Márquez.

**Supervision:** Edna Judith Márquez.

**Validation:** Kevin León García-Castro, Edna Judith Márquez.

**Visualization:** Kevin León García-Castro, Edna Judith Márquez.

**Writing – original draft:** Kevin León García-Castro, Edna Judith Márquez.

**Writing – review & editing:** Kevin León García-Castro, Edna Judith Márquez.

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
