## [Decision Letter · Decision Letter 0]

10 Jan 2024

PONE-D-23-37221Temporal analysis of genetic diversity and gene flow in the threatened catfish Pseudoplatystoma magdaleniatum from a dammed Neotropical RiverPLOS ONE

Dear Dr. Marquez,

Thank you for submitting your manuscript to PLOS ONE. After careful consideration, we feel that it has merit but does not fully meet PLOS ONE’s publication criteria as it currently stands. Therefore, we invite you to submit a revised version of the manuscript that addresses the points raised during the review process.

**I have now received comments from two reviewers on this manuscript. They both agree that the manuscript is technically sound but they both recommended minor revisions to the text before it can be recommended for publication. Reviewer 1 also made edits directly on two pdf files that are attached. I think that the authors should incorporate all of the reviewers' comments as this will greatly improve the quality of the manuscript.**

We look forward to receiving your revised manuscript.

Kind regards,

Jesus E. Maldonado, Ph.D

Academic Editor

PLOS ONE

Journal Requirements:

Whilst you may use any professional scientific editing service of your choice, PLOS has partnered with both American Journal Experts (AJE) and Editage to provide discounted services to PLOS authors. Both organizations have experience helping authors meet PLOS guidelines and can provide language editing, translation, manuscript formatting, and figure formatting to ensure your manuscript meets our submission guidelines. To take advantage of our partnership with AJE, visit the AJE website (http://aje.com/go/plos) for a 15% discount off AJE services. To take advantage of our partnership with Editage, visit the Editage website (www.editage.com) and enter referral code PLOSEDIT for a 15% discount off Editage services. If the PLOS editorial team finds any language issues in text that either AJE or Editage has edited, the service provider will re-edit the text for free.

This study was supported by a grant framed under the Project “Variabilidad genética de un banco de peces de los sectores medio y bajo del río Cauca” (CT-2019-000661, Empresas Públicas de Medellín and Universidad Nacional de Colombia, Sede Medellín). Funders do not play any role in the study design, data collection and analysis, decision to publish, or preparation of the manuscript.

This study was supported by a grant framed under the Project “Variabilidad genética de un banco de peces de los sectores medio y bajo del río Cauca” (CT-2019-000661, Empresas Públicas de Medellín and Universidad Nacional de Colombia, Sede Medellín).

This study was supported by a grant framed under the Project “Variabilidad genética de un banco de peces de los sectores medio y bajo del río Cauca” (CT-2019-000661, Empresas Públicas de Medellín and Universidad Nacional de Colombia, Sede Medellín). Funders do not play any role in the study design, data collection and analysis, decision to publish, or preparation of the manuscript.

5. We note that your Data Availability Statement is currently as follows: All relevant data are within the manuscript and its Supporting Information files.

Additional Editor Comments:

I have now received comments from two reviewers on this manuscript. They both agree that the manuscript is technically sound but they both recommended minor revisions to the text before it can be recommended for publication. Reviewer 1 made edits directly on two pdf files that are attached. I think that the authors should incorporate all of the reviewers' comments as this will greatly improve the quality of the manuscript.

Reviewers' comments:

Reviewer's Responses to Questions

**Comments to the Author**

1. Is the manuscript technically sound, and do the data support the conclusions?

Reviewer #1: Yes

Reviewer #2: Yes

2. Has the statistical analysis been performed appropriately and rigorously? 

Reviewer #1: Yes

Reviewer #2: Yes

3. Have the authors made all data underlying the findings in their manuscript fully available?

Reviewer #1: No

Reviewer #2: Yes

4. Is the manuscript presented in an intelligible fashion and written in standard English?

Reviewer #1: No

Reviewer #2: Yes

5. Review Comments to the Author

Reviewer #1: Striped catfish Psuedoplatystoma magdalineatum is an important, fished species in the Magdalena drainage of Columbia. It has declined to the point that it is listed as endangered. Hence, assessment of population genetic variation is pertinent to gain an understanding of its genetic viability, to gain insights into its recent demographic history, and to delineate populations as units for fishery management, Garcia-Castro and Marquez screened 13 species-specific microsatellite DNA markers and applied a battery of standard population genetic tests to old and new samples collected from the lower Magdalena-Cauca basin. They found considerable genetic diversity, one panmictic population, and evidence of a recent genetic bottleneck, findings which will inform management. The authors appropriately called for population genetic assessment at a larger spatial scale. The study design is straightforward, and the statistical tests and interpretations are appropriate. The findings can be presented more crisply as suggested below. I’ve marked the manuscript to guide revision of the English prose; I hope the authors find my marks constructive.

Methods. – At line 135, I believe the authors do not mention site S1 in error.

Results. – The paragraph starting on line 231 should mention whether any particular loci or populations presented frequent departures from Hardy-Weinberg equilibrium.

At lines 245-246, the authors oversell the likelihood of the infinite alleles model (IAM) applying to their data. Most frequently, it is the two-phase mode that best applies to microsatellite data, often with about 80% stepwise mutation and 20% non-stepwise mutation. Why say anything about the likelihood of the IAM applying best? Why not simply report the results obtained, that IAM yielded a significant test result and the other models did not? The authors have a low M Garza and Williamson (2001) metric, and the cognizant reader will make their own judgement of there having been a recent bottleneck.

At line 261, the authors should strike the word “not”. Infinite estimates of Ne do not provide evidence that the population IS very large.

The authors should not make a big deal over the recent samples having a larger estimated Ne than the old samples. The confidence intervals overlap, which should be mentioned at line 265.

I marked the sentence at line 266 to read of genetically effective migration. Now that I have had an hour to reflect on that, I realize that GeneClass2 does not show genetically effective migration, so the authors should not make that change.

At line 275, I think that the authors intend to write within POPULATIONS. A word seems to be missing.

Discussion. – At line 308, the point of the sentence can be sharpened as: environmental and anthropogenic pressures threaten the demographic and genetic viability of the species, particularly those recognized as having some degree of vulnerability.

At line 326, the change in estimated Ne should be characterized as a non-significant increase. The confidence intervals are large, the estimates are imprecise and rather comparable, so the change should not be oversold.

At line 336, the minimum Ne estimates of 414 SUGGEST LONG-TERM evolutionary risks for this POPULATION OF the species.

References. – The literature citations are reasonably clean, but I did mark a few minor issues.

Reviewer #2: The manuscript reports the genetic diversity and population structure of the endangered fish Pseudoplatystoma magdaleniatum, using microsatellite loci. It is well written and presents data worthy of publication in PlosOne. However, there are some points that must be clarified and others that should be modified to improve the article. See my comments and suggestions below.

Line 39-40. Reword as suggested: basin showed healthy genetics after molecular analyses.

Line 44. If it was previously molecularly analysed, please complete the sentence with this information.

Line 65. Reword as suggested: large migratory fish and the most important fishing resource of Colombia

Line 69-71. All these sentences must be reworded. See my suggestion below.

P. magdaleniatum in the middle and lower sectors.... showed high genetic diversity and absence of populational structuring when analysed by microsatellites (quoted reference).

Line 71-74. Reword the entire paragraph since, similarly, these mentioned species did not show genetic structuring. In addition, data on inbreeding or genetic diversity (e.g. Ra, He, Ho) are not reported for all mentioned species.

Line 97. Change "above" to "mentioned"

Line 104-106. Reword as suggested: … genetic monitoring emerges as an alternative to estimate the threat status and potential changes in genetic diversity of wild populations over time, mainly of those species with some degree of conservation concern and subjected to disturbances in its ecosystem (Laikre et al., 2010; Schwartz et al., 2007).

Line 107-108. Delete “Ex-ante and Ex-post, since this information must be provided in the Material and Methods.

Line 110-11: I suggest rephrase this sentence as suggested below:

The expectations of this work were to find high genetic diversity and no population structuring in P. magdaleniatum of the Cauca River, based on the findings previously reported by...

Line 117. Change "Ex-ante and Ex-post samples" to "sampling"

Line 197 - . Rephrase as suggested below:

To explore genetic changes on a temporal scale of the population of this species, genotypes previously obtained for individuals collected during the years 2010 - 2014 (Ex-ante sample) in sectors of the middle and lower sites of the Magdalena-Cauca basin, using the same set of microsatellite loci and the same methodology for genotyping them (see García-Castro et al., 2021), were included in the analysis, totaling of 311 individuals of P. magdaleniatum.

Line 219-220. Make clear how stuttering effects were corrected.

Line 232-233. Reword as suggested: no locus evidenced deviation from equilibrium in S4-S5 and just one locus (Psm18)…

Line 234. Explain what false positive means in that context.

Line 238. Correct to: to only one locus (Psm24).

Line 315. Please quote some more recent biblio. references on this subject.

Line 358-36. Reword as suggested: In conclusion, our results indicate that P. magdaleniatum has high genetic diversity and is not genetically structured along the Cauca River, although preserving evidence of recent reductions in its population size.

6. PLOS authors have the option to publish the peer review history of their article (what does this mean?). If published, this will include your full peer review and any attached files.

Reviewer #1: No

Reviewer #2: **Yes: **Patrícia Domingues de Freitas

---

## [Author Response · Author response to Decision Letter 0]

23 Feb 2024

We really appreciate the detailed revision of the reviewers and have edited the manuscript following your valuable recommendations, which have led to improve our paper.

Reviewer #1: 

Striped catfish Psuedoplatystoma magdalineatum is an important, fished species in the Magdalena drainage of Columbia. It has declined to the point that it is listed as endangered. Hence, assessment of population genetic variation is pertinent to gain an understanding of its genetic viability, to gain insights into its recent demographic history, and to delineate populations as units for fishery management, Garcia-Castro and Marquez screened 13 species-specific microsatellite DNA markers and applied a battery of standard population genetic tests to old and new samples collected from the lower Magdalena-Cauca basin. They found considerable genetic diversity, one panmictic population, and evidence of a recent genetic bottleneck, findings which will inform management. The authors appropriately called for population genetic assessment at a larger spatial scale. The study design is straightforward, and the statistical tests and interpretations are appropriate. The findings can be presented more crisply as suggested below. I’ve marked the manuscript to guide revision of the English prose; I hope the authors find my marks constructive.

Methods. – At line 135, I believe the authors do not mention site S1 in error.

Our sampling sectors comprise seven sectors S2-S8 according to nomenclature adopted and described in (1) . Now, we replaced “eight” by “seven”.

Results. – The paragraph starting on line 231 should mention whether any particular loci or populations presented frequent departures from Hardy-Weinberg equilibrium.

Done. To clarify, we proofread the idea. Now: “Furthermore, it is worth noting that apart from Psm06 in a single sector, no locus exhibited departure from Hardy-Weinberg equilibrium across population (S1 Table), therefore, the significant values across loci observed in sectors S4-S5 and S6-S7-S8, and in the overall Ex-post sample (P-values of 0.042, 0.023, and 0.007, respectively), may be potentially biased by Fisher's Exact test.”.

We added the respective supporting file as: 

“S1 Table. Hardy-Weinberg equilibrium test per locus and population. P-values in bold (< 0.05) are significant.”

At lines 245-246, the authors oversell the likelihood of the infinite alleles model (IAM) applying to their data. Most frequently, it is the two-phase mode that best applies to microsatellite data, often with about 80% stepwise mutation and 20% non-stepwise mutation. Why say anything about the likelihood of the IAM applying best? Why not simply report the results obtained, that IAM yielded a significant test result and the other models did not? The authors have a low M Garza and Williamson (2001) metric, and the cognizant reader will make their own judgement of there having been a recent bottleneck.

Done. Now: “The result of the excess heterozygosity test to detect recent bottleneck events was significant (P <0.017) in the three sectors and overall (Ex-post) using the IAM, whereas in the other models were non-significant”

At line 261, the authors should strike the word “not”. Infinite estimates of Ne do not provide evidence that the population IS very large.

Done. Now: “Estimates equal to ∞ can be fully explained by a sampling error that is greater than the genetic drift signal, so they do not provide evidence that the population is very large [45] ”.

The authors should not make a big deal over the recent samples having a larger estimated Ne than the old samples. The confidence intervals overlap, which should be mentioned at line 265.

Done. Now: “Therefore, both methods suggest that the Ne of P. magdaleniatum is greater than 1,300 in both temporal samples, noting that confidence intervals overlap, being the lowest limit value of 414 (Table 2).”

I marked the sentence at line 266 to read of genetically effective migration. Now that I have had an hour to reflect on that, I realize that GeneClass2 does not show genetically effective migration, so the authors should not make that change.

Ok.

At line 275, I think that the authors intend to write within POPULATIONS. A word seems to be missing.

Done. Now for clarity: “Although the overall structure index calculated using the AMOVA was significant (F´ST [2, 327] = 0.014; P = 0.013), geographical genetic structure for the Ex-post sample was fully explained by the variances among individuals (6%) and within individuals (94%).”

Discussion. – At line 308, the point of the sentence can be sharpened as: environmental and anthropogenic pressures threaten the demographic and genetic viability of the species, particularly those recognized as having some degree of vulnerability.

Done. Now: “Factors supporting the need for ongoing genetic evaluation of wild populations are related to environmental and anthropogenic pressures that threaten the demographic and genetic viability of the species, particularly those recognized as having some degree of vulnerability [28,29].”

At line 326, the change in estimated Ne should be characterized as a non-significant increase. The confidence intervals are large, the estimates are imprecise and rather comparable, so the change should not be oversold.

Done. Now: “Although some factors such as the presence of individuals of different generations (age structure) or high levels of immigration in the population can bias the estimation and could explain the observed differences between Ex-ante and Ex-post samples [65, 66], such increasing in Ne remains uncertain mainly due to large confidence intervals and the imprecise estimation.”

At line 336, the minimum Ne estimates of 414 SUGGEST LONG-TERM evolutionary risks for this POPULATION OF the species.

Done. Now: “Although the estimates obtained in this study are higher than 1,300, the confidence intervals show the minimum values of 414, suggesting long-term evolutionary risks for the species [67], so this last value must be considered for support management measures to prevent genetic erosion of P. magdaleniatum, considering its current conservation status.”

References. – The literature citations are reasonably clean, but I did mark a few minor issues.

Done.

Reviewer #2: 

The manuscript reports the genetic diversity and population structure of the endangered fish Pseudoplatystoma magdaleniatum, using microsatellite loci. It is well written and presents data worthy of publication in PlosOne. However, there are some points that must be clarified and others that should be modified to improve the article. See my comments and suggestions below.

Line 39-40. Reword as suggested: basin showed healthy genetics after molecular analyses.

Done. Now: “…its population in the lower Magdalena-Cauca basin showed healthy genetics after molecular analyses.”

Line 44. If it was previously molecularly analysed, please complete the sentence with this information.

Done. Now: “This work analyzed a total of 164 samples from the Cauca River collected downstream the Ituango Dam between 2019 – 2021 using species-specific microsatellite markers to compare the genetic diversity and structure in samples collected between 2010 – 2014 from the lower Magdalena-Cauca basin, previously analyzed”

Line 65. Reword as suggested: large migratory fish and the most important fishing resource of Colombia

Done. Now: “One of these endemic species is the striped catfish Pseudoplatystoma magdaleniatum, a large migratory fish and the most important fishery resource of Colombia … ”

Line 69-71. All these sentences must be reworded. See my suggestion below.

P. magdaleniatum in the middle and lower sectors.... showed high genetic diversity and absence of populational structuring when analysed by microsatellites (quoted reference).

Done. Now: “The population of this species in the middle and lower sectors of the Magdalena-Cauca basin showed high genetic diversity and absence of population structure [8]. ”

Line 71-74. Reword the entire paragraph since, similarly, these mentioned species did not show genetic structuring. In addition, data on inbreeding or genetic diversity (e.g. Ra, He, Ho) are not reported for all mentioned species.

Done. Now: “Other migratory species of high commercial interest, such as Pimelodus yuma (nicuro), Pimelodus grosskopfii (barbudo) and Prochilodous magdalenae (bocachico) not only showed gene flow in the middle and lower sectors of the Cauca River but also high degree of inbreeding [3].”

Line 97. Change "above" to "mentioned"

Done. Following the recommendations of both reviewers: “Knowledge about real impacts of such factors on population genetics of non-fragmented species is quite limited.”

Line 104-106. Reword as suggested: … genetic monitoring emerges as an alternative to estimate the threat status and potential changes in genetic diversity of wild populations over time, mainly of those species with some degree of conservation concern and subjected to disturbances in its ecosystem (Laikre et al., 2010; Schwartz et al., 2007).

Done. Now: “…genetic monitoring emerges as an approach to estimate the threat status and potential changes of wild populations over time, mainly of those species with some degree of conservation concern and subjected to disturbances in its ecosystem [28, 29].”

Line 107-108. Delete “Ex-ante and Ex-post, since this information must be provided in the Material and Methods.

Done. Now: “Therefore, this study analyzed the population genetics of P. magdaleniatum on a temporal scale, using samples collected in the Magdalena-Cauca basin before and after the construction of the Ituango Dam, using species-specific microsatellite markers.”

Line 110-11: I suggest rephrase this sentence as suggested below:

The expectations of this work were to find high genetic diversity and no population structuring in P. magdaleniatum of the Cauca River, based on the findings previously reported by...

Done. Now: “The expectations of this work were to find high genetic diversity and no population structuring in P. magdaleniatum of the Cauca River, based on the findings previously reported by García-Castro et al. [8].”

Line 117. Change "Ex-ante and Ex-post samples" to "sampling"

Done. Now: “the short period of time separating samplings before and after the dam construction in relation to the generation length of this species (approximately four years; see [7])”

Line 197 - . Rephrase as suggested below:

To explore genetic changes on a temporal scale of the population of this species, genotypes previously obtained for individuals collected during the years 2010 - 2014 (Ex-ante sample) in sectors of the middle and lower sites of the Magdalena-Cauca basin, using the same set of microsatellite loci and the same methodology for genotyping them (see García-Castro et al., 2021), were included in the analysis, totaling of 311 individuals of P. magdaleniatum.

Done. Now: “To explore genetic changes on a temporal scale for population of this species, genotypes previously obtained for individuals collected during the years 2010 - 2014 (Ex-ante sample) in sectors of the middle and lower sites of the Magdalena-Cauca basin, using the same set of microsatellite loci and the same methodology for genotyping them (see [8]), were included in the analysis, totaling of 311 individuals of P. magdaleniatum.”

Line 219-220. Make clear how stuttering effects were corrected.

Now: “…and stuttering effects were corrected when present based on Micro-Checker suggestions”

Line 232-233. Reword as suggested: no locus evidenced deviation from equilibrium in S4-S5 and just one locus (Psm18)…

For clarifying this idea, we reworded this paragraph as: “Furthermore, it is worth noting that apart from Psm06 in a single sector, no locus exhibited departure from Hardy-Weinberg equilibrium across population (S1 Table), therefore, the significant values across loci observed in sectors S4-S5 and S6-S7-S8, and in the overall Ex-post sample (P-values of 0.042, 0.023, and 0.007, respectively), may be potentially biased by Fisher's Exact test.”

Line 234. Explain what false positive means in that context.

We meant by false positive the likely bias of the Fisher´s method to detect departure of Hardy-Weinberg equilibrium (HWE) in each population, considering that such method is a global test across loci that combines P-values of different tests, assuming independence of loci, and considering that no locus showed consistent departure from HWE across samples.

Line 238. Correct to: to only one locus (Psm24).

Done. The reworded paragraph is described above.

Line 315. Please quote some more recent biblio. references on this subject.

Done. We added a couple of relevant and more recent citations. Now: “The genetic diversity of the Ex-post sample was high, with HE values higher than both the average of Neotropical catfishes (HE: 0.609 ± 0.210; [57]) and values reported in some populations of other species of the genus Pseudoplatystoma [59–64].”

Line 358-36. Reword as suggested: In conclusion, our results indicate that P. magdaleniatum has high genetic diversity and is not genetically structured along the Cauca River, although preserving evidence of recent reductions in its population size.

Done. Now: “In conclusion, the results of this study indicate that P. magdaleniatum has high genetic diversity and is not genetically structured along the Cauca River, although preserving evidence of recent reductions in its population size.”

References:

1. Landínez-García RM, Márquez EJ. Development and characterization of 24 polymorphic microsatellite loci for the freshwater fish Ichthyoelephas longirostris (Characiformes: Prochilodontidae). PeerJ. 2016 Sep 1;4(9):e2419.

---

## [Editor Report · Decision Letter 1]

20 Mar 2024

Temporal analysis of genetic diversity and gene flow in the threatened catfish Pseudoplatystoma magdaleniatum from a dammed Neotropical River

PONE-D-23-37221R1

Dear Dr. Marquez,

We’re pleased to inform you that your manuscript has been judged scientifically suitable for publication and will be formally accepted for publication once it meets all outstanding technical requirements.

An invoice for payment will follow shortly after the formal acceptance. To ensure an efficient process, please log into Editorial Manager at Editorial Manager® , click the 'Update My Information' link at the top of the page, and double check that your user information is up-to-date. If you have any billing related questions, please contact our Author Billing department directly at authorbilling@plos.org.

Kind regards,

Jesus E. Maldonado, Ph.D

Academic Editor

PLOS ONE

Additional Editor Comments (optional):

The authors have adequately addressed all of the reviewers' comments. The manuscript is now much improved and meets the publication criteria in PlosOne.
---

## [Editor Report · Acceptance letter]

29 Mar 2024

PONE-D-23-37221R1 

PLOS ONE

Dear Dr. Marquez, 

I'm pleased to inform you that your manuscript has been deemed suitable for publication in PLOS ONE. Congratulations! Your manuscript is now being handed over to our production team.

Kind regards, 

on behalf of

Dr. Jesus E. Maldonado 

Academic Editor

PLOS ONE